# Efficiently Mitigating Face-Swap-Attacks: Compressed-PRNU Verification With Sub-Zones

**Ali Hassani** [1,*] , **Hafiz Malik** [1] **and Jon Diedrich** [2]

1 Information Systems, Security and Forensics Lab, University of Michigan-Dearborn, Dearborn, MI 48128, USA; hafiz@umich.edu
2 Research and Advanced Engineering, Ford Motor Company, Dearborn, MI 48124, USA; jdiedri5@ford.com
* Correspondence: alihassa@umich.edu

**Abstract:** Face-swap-attacks (FSAs) are a new threat to face recognition systems. FSAs are essentially imperceptible replay-attacks using an injection device and generative networks. By placing the device between the camera and computer device, attackers can present any face as desired. This is particularly potent as it also maintains liveliness features, as it is a sophisticated alternation of a real person, and as it can go undetected by traditional anti-spoofing methods. To address FSAs, this research proposes a noise-verification framework. Even the best generative networks today leave alteration traces in the photo-response noise profile; these are detected by doing a comparison of challenge images against the camera enrollment. This research also introduces compression and sub-zone analysis for efficiency. Benchmarking with open-source tampering-detection algorithms shows the proposed compressed-PRNU verification robustly verifies facial-image authenticity while being significantly faster. This demonstrates a novel efficiency for mitigating face-swap-attacks, including denial-of-service attacks.

**Keywords:** digital cameras; forensics; face recognition; real-time systems; compressed sensing

## 1. Introduction

Face-recognition (FR) enables users to simply look at the camera and be authenticated. This seamless functionality, however, is reliant upon the authentication being secure. In principle, this is the case. Advances in deep learning have brought precise identification algorithms, capable of discerning one person from over 50,000 [1]. Protection against imposters is now also an industry standard [2]; most applications have the capability to discern live people from pictures, videos and simple masks [1]. A new problem, however, is the photo-realistic face-swap-attack (FSA). Through generative adversarial networks, attackers can imperceptibly present any face desired [3].

To appreciate the risk of FSAs, consider the FR threat model in Figure 1. Historically, the primary FR imposter-attack is presenting a facsimile (scenario 2). To address this, many FR applications now include liveliness-verification technologies. This can include 3D sensing [4], analyzing physiological motions (blink [5], respiration [6] and pulse [7]) and now spatio-temporal fusion [8]. While still an on-going issue, there is general defense against spoofing attacks. Swapping attacks (scenario 3), however, have the potential to bypass these methods. State-of-the-art algorithms imperceptibly align and blend the swapped face, retaining fundamental liveliness features. This means a FSA can be an extremely potent means to gain access to phones [9], ATMs [10], buildings [11] and even vehicles [12].

FSAs also provide the benefit of discretion. If an imposter tries to spoof an ATM or building, bystanders could clearly notice someone wearing a mask. They can attempt to perform the spoofing attack when others are not around, but this would clearly limit their opportunistic window. Conversely, the opportunistic window could instead be used to

place an interception device. From there, they can discretely authenticate at any future point in time.

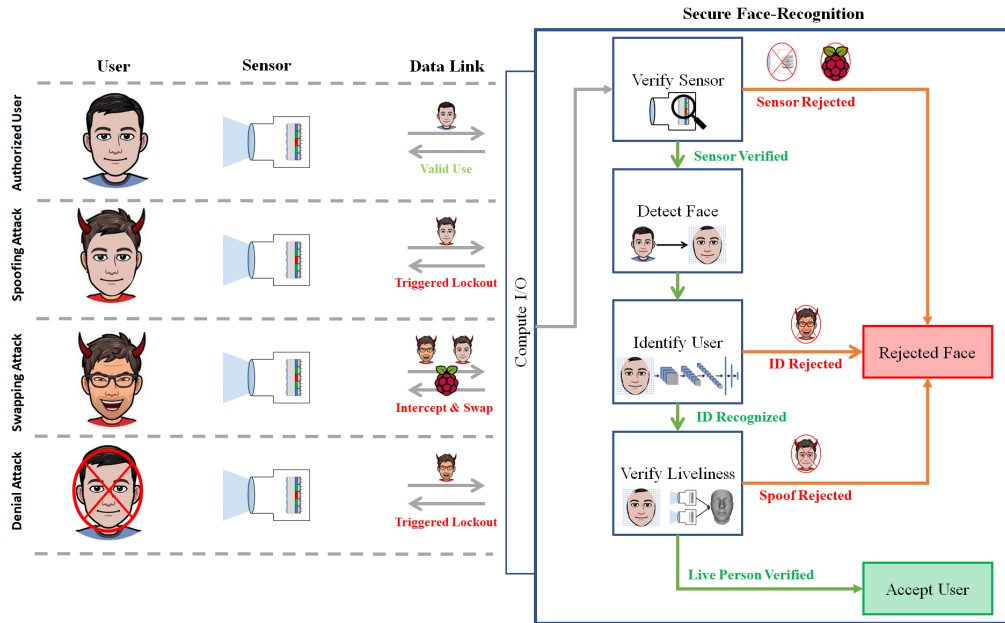

**Figure 1.** Secure face recognition pipeline with attack vectors. Scenario 1 is authenticating an authorized user. Scenario 2 is spoofing via a physical facsimile. Scenario 3 is spoofing via face-swap-attack. Scenario 4 is denying service by destructively altering faces.

Lastly, FSAs are a unique way to deny service (scenario 4). Simplistic service denial would be to imperceptibly alter faces to prevent authentication. A malicious use case would be to avoid surveillance, such as electronic border control. Regardless of how well camera arrays are planned out, using deep learning to present another person (or remove their presence) would go undetected today.

These problems are exacerbated by the ease of success. Many FR systems only use message authentication (and not encryption) to reduce overhead [13]. This inherently makes the data vulnerable to tampering. Furthermore, quality face-swapping tools are ubiquitous. Social media platforms include applications to photo-realistically swap faces and backgrounds [14,15]. This lack of encryption, combined with accessible tools, means any hacker with an interception device can successfully apply FSAs.

Hence, the purpose of this research is to develop an efficient authenticity-verification framework for real-time FSA mitigation. A noise-verification framework is presented to detect changes in sensor noise caused by swapping (in this case photo-response non-uniformity, PRNU) [16]. PRNU is historically known to be useful for tampering analysis via anomaly detection methods [16–18]. This framework instead proposes verification against a compressed enrollment for speed. Given most FR systems use a known sensor, it can be securely characterized for authentic PRNU. Each following frame is then compressed and analyzed for authenticity using peak correlation energy deviation. Local sensitivity is provided through the use of sub-zones.

*Face-Swap-Attack Contributions*

This research presents a novel compressed noise-verification framework to mitigate FSAs. The source camera is characterized for PRNU, where subsequent frames are verified for authenticity. Face-swap-attacks are mitigated by analyzing only the zone containing the face centroid; service-denial-attacks require full image analysis. Benchmarking against three open-source algorithms shows the proposed framework is both robust and significantly more efficient than existing solutions. In summary, this research presents the following contributions:

1. Extremely efficient face-crop verification (93.5% accuracy, 4.6 ms on CPU)
2. Robust full-image verification (100% accuracy, 106.9 ms on CPU)

## 2. Related Works

Detecting face-swap-attacks (FSAs) starts with traditional image tampering detection. In his Image Forgery Detection survey, Dr. Hany Farid presented how images are commonly tampered, along with corresponding detection methods [19]. Images can be tampered for various reasons; people have been known to present fake imagery to forge alibis [20], steal identities [21] or create compromising material [22]. This can be accomplished using a variety of hand-crafted tools, such as: cloning, duplicating parts of the image to conceal or embed information, re-sampling, adapting the resolution to modify scene proportions and splicing, combining multiple images together to create a new scene [19].

### 2.1. Forensics-Based Tampering Detection

To detect these image attacks, a forensics approach can be taken using noise artifacts. Figure 2 shows some of the fundamental components to a camera, where manufacturing tolerances make it nearly impossible to create a perfectly noiseless component. These artifacts can be quantified into a robust "noise fingerprint" or "noiseprint" [19].

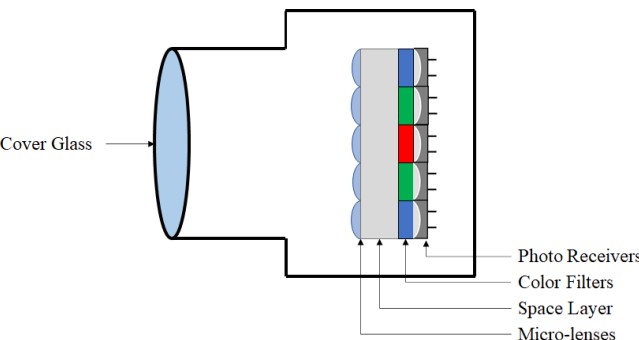

**Figure 2.** Camera components.

One particularly famous "noiseprint" is photo-response non-uniformity (PRNU). This is an estimation of photo receiver noise with respect to a constant uniform light [19] as shown in Figure 3. PRNU is known to be unique across cameras, as validated in the large-scale source identification from Goljan et al. [23]. This uniqueness means that deviations in PRNU can be used to detect tampering. Traditional tampering (e.g., cut-and-splice) has been successfully detected with PRNU anomaly detection techniques [16,17] as well as deep learning networks trained on PRNU [24,25]. The logical question is whether noise analysis can identify deep-learning-based tampering (e.g., deepfakes).

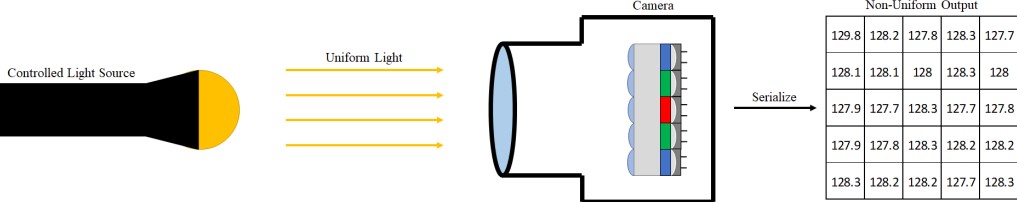

**Figure 3.** Visualizing photo-response non-uniformity. Photo receivers are imperfect due to manufacturing tolerances. When a uniform light is presented to the camera, a unique, non-uniform response is output.

### 2.2. Deep-Learning Tampering Detection

Deepfake detectors often focus on fake videos. Videos incorporate temporal information, whereby it is inherently harder to replicate realistic movements over extended periods.

Guera et al. demonstrated very strong performance, hitting 96% accuracy on test videos when using 20 frames [26]. Others have similarly pursued multi-frame analysis. Sabir et al. demonstrated that strong performance can be achieved with fewer frames if aligning the faces [27]; however, Tariq et al. noted large-scale robustness over varied perspectives does benefit from increasing the frame count (in this case 16) [28]. This approach seems promising, but is inherently not ideal for real-time applications due to the latency in acquiring numerous frames.

To avoid this latency, researchers postulate that face-swap generative adversarial networks (GANs) leave signature traces. Rossler et al., developers of the famous Face-Forensics++ data-set [29], demonstrated performance across a slew of convolutional neural networks. Their network, XceptionNet, performs very well (99% authenticity verification correctness on raw images, 81% on compressed) [29]. This has inspired numerous other high-performing models [30], where some introduce multi-stream fusion [31], multi-task learning [32] and optical flow [33].

### *2.3. Literature Opportunity*

All of these approaches, however, have a fundamental constraint. Each detector must be trained on the specific face-swap GAN. When new GANs are developed, the detector must be re-trained on their signature traces or risk being vulnerable. The ultimate goal is to find use a feature that is invariant to the tampering method and can be calculated in real time. This brings a return to fundamental camera forensics. Concurrent to this research, researchers are demonstrating face-swaps similarly present anomalies in PRNU. Lugstein et al. published that PRNU spectral trace analysis is a reliable feature [18]. They achieved between 95% and 99% in authenticity verification across a multitude of data-sets [18]. Mohanty also corroborated this methodology by presenting a high-level framework for verifying PRNU against enrolled templates [34].

This research provides novelty over that of Lugstein and Mohanty by incorporating compressed zonal analysis in the PRNU verification. The introduction of compression is important for run-time latency; this algorithm needs to be extremely efficient or it will not be deployed on any of the real-time applications from the introduction. The compression, however, inherently reduces feature precision. The zonal analysis provides local sensitivity to retain robustness. Lastly, it is key to note that both papers are published concurrently to this research; no prior knowledge of their methods is used here.

In summary, this framework contributes through state-of-the-art efficiency. PRNU analysis is clearly becoming a respected method for determining deep-network manipulations. This research extends the field by introducing compressed verification with zonal analysis. This enables best-in-class performance in a fraction of the run-time.

### 3. Materials and Methods

Image tampering is broadly defined as any digital manipulation of the original source. The fundamental hypothesis is that any image modification inherently changes the underlying "noiseprint" (e.g., PRNU); hence, this methodology uses deviation in the expected "noiseprint" to detect and localize tampered segment(s).

### *3.1. Photo-Response Non-Uniformity Estimation*

The PRNU calculation employs the methodology presented by Goljan et al. [23]. In general, an image can be described as the sum of the incident light received by the camera, artifacts introduced by camera intrinsics and temporal noise. This is expressed in Equation (1), where the incident light is represented by $I_0$, the camera noise (PRNU) is represented by $K$ and other noises (quantization, shot, dark current, temporal, etc.) are represented by $\theta$:

$$I = I_0 + I_0K + \Theta \tag{1}$$

To isolate the "noiseprint", the image must be first filtered to remove noise $\theta$. This can be accomplished by applying a Wiener filter, $F$, to generate residuals $W^i = I^i - F(I^i)$ [35].

Note that dark current is assumed to be negligible due to having sufficient scene signal. From there, $K$ can be isolated by applying a maximum likelihood estimator [35]:

$$\hat{K} = \frac{\sum_{i=1}^{N} W^i I^i}{\sum_{i=1}^{N} (I^i)^2} \tag{2}$$

### 3.2. Peak Correlation Energy

One way to classify the camera noise source is to utilize peak correlation energy (PCE). The PCE value is computed using the MATLAB code provided by Goljan et al. [23]. This approach calculates the Pearson correlation coefficient and then identifies the maximal value for a given sliding window [23]. This is achieved by first computing the noise residuals of the hypothesis camera, $X$, and the image, $Y$, as a sum of PRNU, $K$ and secondary noises $\Theta$:

$$\begin{aligned} X &= I\hat{K} \\ Y &= IK_{image} + \Theta_{image} \end{aligned} \tag{3}$$

The correlation is computed over shifted areas, where the maximal number of shifts is defined as the product of the difference in image $m \times n$ versus fingerprint dimensions $m_k \times n_k$: $max = (m_k - m + 1)(n_k - n + 1)$ [23] , that is:

$$\begin{aligned} \rho(s_1, s_2; X, Y) &= \\ \frac{\sum_{k=1}^{m} \sum_{k=1}^{n} (X[k,l] - \bar{X})(Y[k+s_1, l+s_2] - \bar{Y})}{|X - \bar{X}||Y - \bar{Y}|} \end{aligned} \tag{4}$$

By definition, this implies that $PCE$ is the correlation value for which the peak occurred for shift vector, $s_{peak} = [s_{1max}, s_{2max}]$. In [23], Goljan et al. suggested having the local peak area, $\eta$, to be an $11 \times 11$ pixel grid. This is given in Equation (5):

$$PCE_k(X, Y) = \frac{\left(X \cdot Y(s_{peak})\right)^2}{\frac{1}{mn - |\eta|} \sum_{(s_1, s_2) \in \eta} (X \cdot Y(s_1, s_2))^2} \tag{5}$$

### 3.3. Tampering Score: Face-Swap Verification

Detected faces can be directly verified for tampering. This is accomplished by identifying face centroid's zone, $z$, and applying a tampering score on it. Tampering score is generated by applying a tampering filter on the correlation energy. The high-pass cutoff is calibrated to ignore standard noise as tampering; the low-pass cutoff is calibrated to ignore images that do not match the source. This calibration is accomplished on a per-camera basis; in this case, a 1% false acceptance rate is prioritized. This face-zone-verification (FZV) score is described in Equation (6):

$$T_{FZV} = BP_z(PCE_K(X_z, Y_z)) \tag{6}$$

The FZV score offers a significant run-time improvement at the cost of some robustness (as only the facial pixels are evaluated pixels). In theory, this can be expanded to a nearest neighbor approach and also to evaluate all zones directly adjacent to the centroid zone $z$.

$$T_{DZV} = BP_z(K_{rc}), rc \in Z \tag{7}$$

### 3.4. Tampering Score: Service-Denial Verification

This tampering score can be used to verify full image integrity in a zonal-expected-value (ZEV) fashion to mitigate face removal. The tampering filter can be applied on each zone, designated $BP_z$, where a final filter is applied on the averaged output, $BP_I$. Each zone

filter can be individually calibrated, though for pragmatism uniform cutoffs are assumed. The complete filtered score is described in Equation (8):

$$T_{ZEV} = BP_I \left( \frac{1}{Z} \times \sum_{r,c \in Z} BP_z(PCE_K(X_{rc}, Y_{rc})) \right) \tag{8}$$

If on average the image appears tampered across zones, it is then labeled as tampered. Otherwise, it is labeled authentic or different source, based upon whether it fails the high-pass or low-pass cut off, respectively.

### 3.5. Compression via Down-Sampling

Compression is introduced in the form of down-sampling to optimize run-time performance. Down-sampling is chosen because it is computationally cheap and implicitly behaves as an averaging filter when combining neighboring pixels. For reference, the enrollment image is extracted for PRNU at full scale and then down-sampled to maximally retain features; challenge images, however, are down-sampled first then extracted for PRNU to minimize run-time.

All images are evaluated using three compression settings. These are enumerated below, with the axis sampling rate provided in the form of (row, column):

1. Full-Scale Resolution (1 × 1)
2. Quarter-Scale Resolution (1/2 × 1/2)
3. Sixteenth-Scale Resolution (1/4 × 1/4)

### 4. Experimental Setup

The proposed framework is evaluated on a constructed data-set. This route is chosen in lieu of using a public data-set because the verification methodology requires knowledge of the source camera, with both authentic and tampered images. Hence, facial images are acquired using 3 Ras Pi cameras. This maintains same scene with same imaging sensor. This approach is used to minimize differences across images, e.g., when image regions are swapped between cameras, it should appear visibly identical. Note these experiments are for facial image authenticity verification, but for completeness a sensitivity analysis of compressed PRNU in source identification is also provided in the appendix (Appendix A). In all cases, run-time is evaluated to demonstrate real-time applicability on a Dell Latitude E5570 laptop (single CPU core only). For simplicity, it is assumed that the image is already read into memory where only the PRNU calculation and the camera classification time are measured.

### 4.1. Exp 1: Direct Face-Swap Verification

Exp 1 evaluates where optimizations can be conducted for real-time face recognition by applying face-zone-verification (FZV). That is to say, in the images containing a face, the tampering score is only calculated on the zone containing the face's centroid.

This experiment augments the Ras Pi Camera data-set by utilizing the same cameras to acquire 50 photos containing face-swap volunteers (per camera). Swaps are conducted in manual and artificial intelligence fashions. Manual swap is conducted by identifying the boundary of the detected face in both images and swapping them (resizing the face as appropriate). The artificial intelligence approach employs a well-known online tool, Reflect [36], to do a landmark alignment and merge the faces together.

Face-swaps are conducted using faces from both the same camera and different cameras. This results in 20 swaps generated per swap use case, sampling rate and zone, for a total of 1440 tampered images generated. Figure 4 shows an example of how this is performed using two test subjects. For reference, the face-swap regions range from $\frac{1}{20}$ to $\frac{1}{16}$ image width. This data-set is designated Ras Pi—Face Swap.

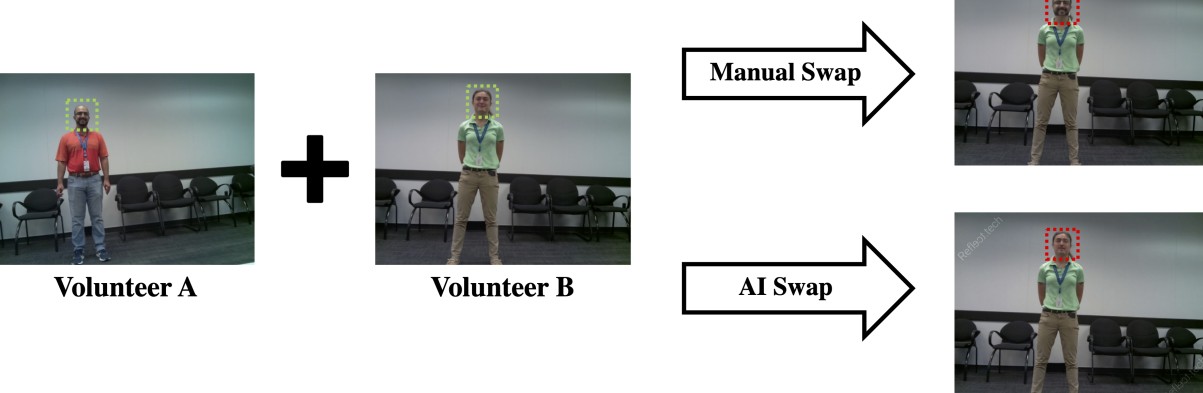

**Figure 4.** Face swap methodology example. Images are acquired from various people using the same background. Swaps are then performed by manually cropping faces and intelligently blending them using an AI tool.

The FZV tampering score is then calculated for the zone containing the centroid of the detected face in each image. Security is also evaluated in an end-to-end perspective, whereby a simple face recognizer is leveraged to validate overall probability of spoofing. The face recognizer uses the histogram of oriented gradients (HOG) feature with a multi-layer perceptron (10 hidden nodes) to represent constrained, real-time applications.

### 4.2. Exp 2: Simulated Service-Denial Verification

After verifying that compressed PRNU can sufficiently verify face integrity, Exp 2 evaluates a simulation of service denial, where faces are removed (i.e., there is no face to detect). Worst-case analysis is conducted by tampering images from the Ras Pi Camera data-set (i.e., all cameras of the same model, where images are of matching scenes).

Face removal across the image is simulated by randomly swapping blobs using matching scenes across images. That is to say, a blob of matching background is placed, mimicking the effect of removing faces. The swap shape chosen is a circle to mimic the shape of a face. A swap example is shown in Figure 5; note that the red outlines shown only identify the swap region and are not present in the actual tampered images.

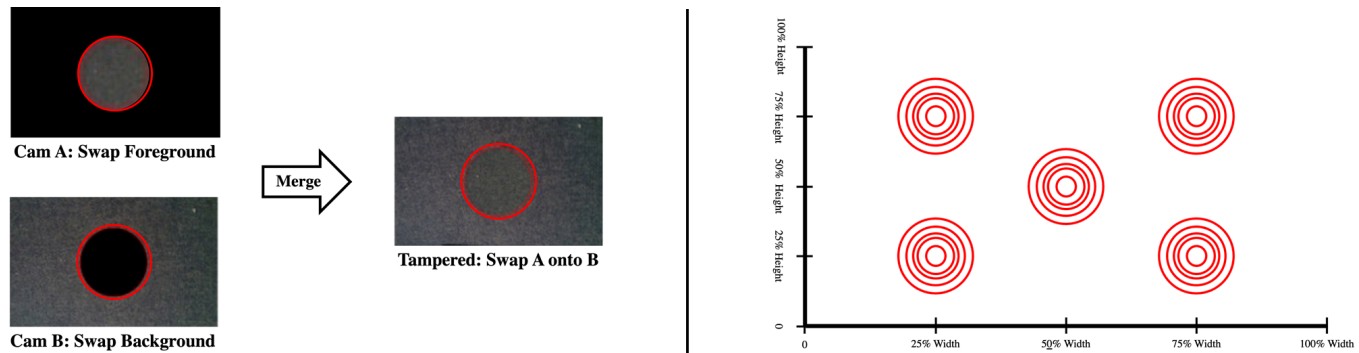

**Figure 5.** Image blob-swap methodology example with blob-swap locations. Blobs are randomly exchanged across images to evaluate the tampering score sensitivity to size and position.

Swaps are performed in the image center, top left quadrant, top right quadrant, bottom left quadrant and bottom right quadrant. Swap size is varied by taking radii from $\frac{1}{12}$th image width to $\frac{1}{50}$th image width. These locations are indicated in Figure 5. For reference, this translates to a worst-case radius range of 9 to 40 pixels at the sixteenth down-sample condition.

This generates 20 images per swap location, per sample rate for a total of 1500 tampered images. Tampering score is evaluated for all images by comparing the tampered image

against the expected source. Performance is defined as the mean average precision of labeling images as tampered or authentic. This data-set is designated Ras Pi—General Swap.

The ZEV tampering score is then calculated by computing an expected value across all zones in the image. Again, the purpose here is to simulate the detectability of face deletion across image locations (via matching blob-swaps).

### 4.3. Open-Source Benchmark Algorithms

The noise-verification framework is benchmarked against three open-source algorithms. The algorithms are selected because they are the best performing relevant algorithms on GitHub. Note that the FZV and ZEV algorithms are written in MATLAB; however, the open-source algorithms are found written in Python. For pragmatic reasons, the code is used as is. This has no bearing on accuracy, but it gives a slight run-time evaluation to the proposed methods (MATLAB is optimized for matrix multiplication).

The first algorithm is a deep learning approach leveraging error level analysis (ELA) [37]. Gunawan et al. developed this approach by transforming images using ELA and feeding a lightweight, convolutional neural network examples of authentic and spliced images from an open-source tampering data-set [38]. For this evaluation, the network is re-trained on the same experimental data. This algorithm is referred to as "ELA CNN".

The second algorithm comes from an ensemble image forgery detection toolbox [39]. Levandoski and Lobo presented deterministic algorithms for re-compression, color filter array anomalies, noise variance and generic image duplication due to copying [39]. It is key to note that due to run-time issues, the color filter array analysis is omitted as the algorithm takes over 30 min per image on this research machine and is not viewed as real-time. Given there that is no calibration setting, the algorithm suite is run directly on the experimental data without modification. This algorithm is referred to as "Forgery Tool".

The third algorithm is discrete wavelet transform analysis for blind image tampering [40]. While in a slightly older paper (2009), Mahdian and Saic applied a methodology that aligns very similarly with this paper's PRNU analysis, the key novelty is the inclusion of a priori camera knowledge. For this evaluation, a tampering threshold is calibrated on the same experimental data. This algorithm is referred to as "DWT".

### 4.4. Research Limitations

It is acknowledged that this verification approach does require a priori knowledge of the camera intrinsics. Furthermore, it is relevant to note there are real-world noise factors that this study did not introduce. For example, social media applications may employ their own proprietary compression methods to streamline data transmission; Meij et al. have started this investigation specifically for the WhatsApp communication tool [41], noting some degradation is to be expected. Additionally, real-world camera exterior camera applications, such as security monitoring, introduce environmental noises (e.g., ambient light, rain, dirt, dust, heat, etc.). Validating the impact of the noises would further justify this methodology as well as potentially identify other novel control methods.

## 5. Results

The two experiments are evaluated to validate the tampering score algorithms (against face-swaps and randomized blob-swaps). Classification performance is presented in the form of mean average precision in authenticity prediction. Analysis of down-sampling impact on PRNU features can be found in the appendix (Appendix A).

### 5.1. Exp 1: Direct Face-Swap Verification

The first experiment evaluates whether a detected face can be directly verified for authenticity. The face-zone verification (FZV) algorithm is benchmarked on the Ras Pi— Face-swap data-set (1440 images constructed from no-swap, manual-swap and AI face-swaps [36]). The FZV tampering score is evaluated only on the zone containing the face

centroid, using 1, 16 and 100 zones. Face centroid zone is estimated using the MATLAB cascade object detector [42].

Table 1 presents the benchmark face-swap-attack verification performance for the FZV and open-source algorithms. Authentic (no swap) image is represented by "Authentic". For readability, the results table is simplified to represent all manual face-swap-attacks as "Manual Face-Swap" and all AI face-swap-attacks as "AI Face-Swap". A full sensitivity analysis for the FZV algorithm can be seen in the appendix (Appendix B), verifying the difference between same-camera and different-camera swap performance is small.

**Table 1.** Face-swap-attack verification performance.

| Image Tampering | FZV 1 Zone (Accuracy) | FZV 16 Zones (Accuracy) | FZV 100 Zones (Accuracy) | ELA CNN (Accuracy) | Forgery Tool (Accuracy) | DWT (Accuracy) |
|---|---|---|---|---|---|---|
| Authentic Full-Scale | 100% | 100% | 100% | 91.3% | 98.8% | 26.2% |
| Manual Face-Swap Full-Scale | 100% | 100% | 100% | 100% | 99.0% | 100% |
| AI Face-Swap Full-Scale | 100% | 100% | 100% | 100% | 99.0% | 100% |
| **Full-Scale Mean** | **100%** | **100%** | **100%** | **98.3%** | **99.8%** | **85.3%** |
| Authentic Quarter-Scale | 92.5% | 92.5% | 92.5% | 48.8% | 30.4% | 1.3% |
| Manual Face-Swap Quarter-Scale | 100% | 100% | 92.5% | 57.5% | 100% | 100% |
| AI Face-Swap Quarter-Scale | 91.3% | 87.5% | 45.0% | 53.8% | 100% | 98.8% |
| **Quarter-Scale Mean** | **95.0%** | **93.5% †** | **73.5%** | **54.3%** | **74.8%** | **79.8%** |
| Authentic Sixteenth-Scale | 85.0% | 85.0% | 85.0% | 45.0% | 0.0% | 0.0% |
| Manual Face-Swap Sixteenth-Scale | 81.8% | 81.8% | 77.5% | 51.8% | 100% | 100% |
| AI Face-Swap Sixteenth-Scale | 81.3% | 81.3% | 27.5% | 51.3% | 100% | 94.4% |
| **Sixteenth-Scale Mean** | **82.0%** | **82.0%** | **59.0%** | **50.2%** | **80.0%** | **77.8%** |

† represents the optimized face-swap detection algorithm: FZV with 16 zones at quarter-scale.

These results demonstrate the FZV approach generally outperforms the open-source algorithms for mitigating face-swap-attacks. In particular, the FZV algorithm is optimized at quarter-scale with 16 sub-zones (indicated by the †). This includes 100% accuracy with manual FSAs and general robustness to AI FSAs in an efficient manner (run-time is shown in Section 5.3). Counter to intuition, the hypothesis of isolating the relevant pixels for noise analysis seems to reduce performance. The small number of pixels instead seems to produce a less reliable PRNU measurement, impacting tampering score precision. One possible way to improve performance with reasonable overhead is to apply a nearest neighbor approach, including all adjacent zones.

The open-source results reinforce that the AI face-swap detection is a significant challenge. At full-scale resolution, all three perform well. However, the aggressive calibration becomes problematic for down-sampled imagery. The deep learning approach (ELA CNN) [37] degrades to approximately a coin flip; the two deterministic algorithms [39,40] effectively determine every image to be tampered (arguably worse). This shows the utility of using a noise-verification approach in lieu of traditional classification.

In all cases, the compressed AI face-swaps are difficult to precisely verify. Two hypotheses are proposed. First, fewer pixels are tampered due to the intelligent blending. Second, the landmark-merged face does not resemble the target enough to fool a reasonable face recognizer. The first hypothesis is challenging to mitigate; however, the second implies the identification algorithm can provide end-to-end security. Note that the actual identification would be conducted at full-scale resolution. The benefit of the noise-verification framework is to only perform the authenticity analysis at compression; this means the full precision of the identification algorithm can be utilized. Appendix B verifies that only detectable face-swaps pass an implemented face recognition test.

### 5.2. Exp 2: Simulated Service-Denial Verification

The second experiment evaluates the detectability of simulated service-denial-attack. Service-denial-attacks imperceptibly remove faces, meaning the full image needs to be verified for authenticity. To this end, blob-swaps are performed for the sizes and locations shown in Figure 5. The full-image, zonal-expected-value (ZEV) algorithm is evaluated on the Ras Pi—General Swap data-set (1500 images constructed from no-swap and blob-swap images), using 1, 16 and 100 zones.

Table 2 presents the benchmark service-denial-attack verification performance for the ZEV and open-source algorithms. Authentic (no swap) images are represented as "Authentic". For readability, the results table is simplified to represent blob swap locations as "Service-Denial". A full sensitivity analysis for the ZEV algorithm can be seen in the appendix (Appendix B), verifying the difference across location is small.

**Table 2.** Simulated service-denial verification performance.

| Image Tampering | ZEV 1 Zone (Accuracy) | ZEV 16 Zones (Accuracy) | ZEV 100 Zones (Accuracy) | ELA CNN (Accuracy) | Forgery Tool (Accuracy) | DWT (Accuracy) |
|---|---|---|---|---|---|---|
| Authentic Full-Scale | 100% | 100% | 100% | 99.7% | 99.0% | 70.2% |
| Service-Denial-Attack Full-Scale | 100% | 100% | 100% | 99.8% | 99.0% | 41.3% |
| **Full-Scale Mean** | **100%** | **100%** | **100%** | **99.8%** | **99.0%** | **46.1%** |
| Authentic Quarter-Scale | 92.5% | 100% | 100% | 54.8% | 90.3% | 59.5% |
| Service-Denial Quarter-Scale | 82.3% | 98.8% | 100% | 64.7% | 70.0% | 34.9% |
| **Quarter-Scale Mean** | **84.0%** | **99.0%** | **100%** | **63.0%** | **73.3%** | **39.0%** |
| Authentic Sixteenth-Scale | 85.0% | 92.5% | 100% | 40.2% | 26.1% | 55.0% |
| Service-Denial Sixteenth-Scale | 74.4% | 89.1% | 100% | 59.9% | 0.1% | 16.0% |
| **Sixteenth-Scale Mean** | **74.4%** | **89.0%** | **100% ‡** | **56.6%** | **12.7%** | **9.3%** |

‡ represents the optimized service-denial detection algorithm: ZEV with 100 zones at sixteenth-scale.

The results demonstrate the ZEV approach also generally outperforms the open-source algorithms for mitigating service-denial-attacks. The ZEV optimizes performance at sixteenth-scale with 100 sub-zones (indicated by the ‡). 100% accuracy is achieved over all blob-swaps and authentic images with relative efficiency (run-time shown in Section 5.3). This holds true over all blob swap sizes and locations, which are as small as $\frac{1}{50}$th image-width. The utility of the zonal analysis really presents itself here, where significant compression can be applied while retaining full robustness.

The open-source algorithms conversely show significant degradation with slight compression. The deep learning algorithm again seems to stabilize at a coin-flip, but generally speaking all are unreliable under heavy compression. There is at least less over-fitting this time, showing a smaller difference between authentic and blob-swap images. This is postulated to be a result of the very small swaps, where the lack of local analysis significantly deteriorates performances. This further validates the utility of the proposed framework when using compression.

### 5.3. Run-Time Analysis

Authenticity-verification algorithms must fit within the primary application latency requirements (e.g., facial recognition). This is particularly true when considering high-risk use cases, where sensor encryption is minimized to reduce time. To evaluate these impacts, Table 3 quantifies the run-times of face-swap-attack and service-denial-attack verification. The noise-verification framework (NVF) algorithms (FZV and ZEV respectively) are both referred to as NVF here.

**Table 3.** Image tampering detection run-time.

| Image Tampering | NVF 1 Zone (ms) | NVF 16 Zones (ms) | NVF 100 Zones (ms) | ELA CNN (ms) | Forgery Tool (ms) | DWT (ms) |
|---|---|---|---|---|---|---|
| Face-Swap-Attack Full-Scale | 384.1 | 40.0 | 14.7 | 247.0 | 207.9 | 48.2 |
| Face-Swap-Attack Quarter-Scale | 90.2 | **4.6 †** | 2.0 | 206.4 | 67.3 | 9.1 |
| Face-Swap-Attack Sixteenth-Scale | 22.8 | 2.0 | 1.0 | 181.3 | 4.7 | 5.7 |
| Service-Denial-Attack Full-Scale | 507.7 | 537.6 | 1139.6 | 225.6 | 193.8 | 46.3 |
| Service-Denial-Attack Quarter-Scale | 97.3 | 103.9 | 214.3 | 192.6 | 67.9 | 9.9 |
| Service-Denial-Attack Sixteenth-Scale | 33.9 | 50.2 | **106.9 ‡** | 168.8 | 4.8 | 5.3 |

† represents the optimized face-swap detection algorithm: FZV with 16 zones at quarter-scale. ‡ represents the optimized service-denial detection algorithm: ZEV with 100 zones at sixteenth-scale.

Table 3 benchmarks the run-time of the proposed and open-source algorithms. The optimized ZFZ algorithm is indicated by † and the optimized ZEV algorithm is indicated by ‡. These results demonstrate the noise-verification framework can provide a notable advantage in efficiency. When it comes to face-swap-attack mitigation, the optimal open-source algorithm would be DWT at full-scale. This takes 48.2 ms, in comparison to the ZEF taking 4.6 ms. This shows a substantial improvement, where the ZEF approach is ideal for doing per-frame analysis with negligible overhead.

Service-denial-attack mitigation is inherently more computationally expensive. Rather than just verifying the detected face, the full image must be verified for authenticity (e.g., face removal). Here, the ZEV algorithm again shows notable advantages in efficiency. The optimal open-source algorithm is the Forgery Tool at full scale, which takes 193.8 ms in compared to the ZEV's 106.9 ms. While an improvement, this is still too slow for per-frame verification. Instead, it is suggested to do a periodic full-image authenticity challenge to verify a denial attack is not present.

## 6. Conclusions

In summary, this research presents a novel way to efficiently mitigate face-swap-attacks. Similar to spoofing attacks, face-swap-attacks enable imposters to fool authentication systems and avoid surveillance. Here, a noise-verification framework is presented to robustly verify facial image authenticity with minimal overhead. This is achieved by first characterizing the source camera for PRNU and then evaluating deviations from future images under compression. Experimental results show the proposed framework is both robust and significantly faster than the benchmarked algorithms.

The face-zone-verification algorithm can robustly mitigate face-swap-attacks. One hundred percent of images are correctly predicted as authentic and tampered at full-scale. Performance is then optimized by applying quarter-scale down-sampling with 16 sub-zones; this achieves 93.5% accuracy while taking only 4.6 ms on CPU (a 99.1% reduction). In comparison, the optimal benchmark algorithm takes 48.2 ms—a significant reduction. This is fast enough to be conducted on a per-frame basis with negligible overhead to the face recognition application. If greater security is necessary, the full-scale imagery with 100 sub-zones achieves 100% verification accuracy with a small increase in run-time.

To mitigate service-denial by removing faces (e.g., surveillance avoidance), a periodic zonal-expected-value challenge is also recommended. The optimized algorithm (sixteenth-scale with 100 sub-zones) achieves 100.0% verification accuracy while taking 106.9 ms. While notably faster than the optimal benchmark algorithm (193.8 ms), this is still too slow for a per-frame analysis. Instead, this is best served as a periodic security challenge. It suggests the best times to do the ZEV algorithm are upon application initialization and whenever no face is detected for extended periods.

*Future Directions*

Our future research aims to investigate if the developed models can be generalized to no longer require a priori knowledge of the camera. The deviation in "noiseprint" that is caused by performing swaps or blending images together could be identified via anomaly detection methods. This approach would likely require deep learning, where it is likely that an AI system-on-chip would be necessary for real-time deployment. With this said, naive PRNU-based tampering detection would have tremendous value in media distribution.

## 7. Patents

This research has generated patent applications jointly filed between Ford Motor Company and the University of Michigan. If allowed, a patent number is provided; those that are still in process are identified by case ID.

1. VISION SENSOR DYNAMIC WATERMARKING VIA NOISE CHARACTERIZATION (Issued Patent ID: US11127104). Dynamic watermarking encoded using camera noise as a means to be imperceptible.
2. CAMERA IDENTIFICATION (USPTO Case ID: 84215571US01). A method for efficiently verifying camera source image authenticity using compressed, zonal "noiseprint" analysis.
3. CAMERA TAMPERING DETECTION (USPTO Case ID: 84215575US01). A method for efficiently verifying image authenticity using deviation from expected compressed "noiseprint".
4. CAMERA TAMPERING DETECTION (USPTO Case ID: 84215579US01). A method for optimizing image authenticity verification by only analyzing detected objects.

**Author Contributions:** Conceptualization, A.H. and J.D.; methodology, A.H. and H.M.; software, A.H.; validation, A.H. and J.D.; formal analysis, A.H.; investigation, H.M.; resources, J.D.; data curation, A.H.; writing—original draft preparation, A.H.; writing—review and editing, H.M.; visualization, A.H.; supervision, H.M. and J.D.; project administration, J.D.; funding acquisition, J.D. All authors have read and agreed to the published version of the manuscript.

**Funding:** The authors would like to give special thanks to Ford Motor Company for funding this research via University Alliance Grant, Biometric Forensics.

**Institutional Review Board Statement:** The University of Michigan IBR approved using facial images for face-swap-attack research.

**Informed Consent Statement:** Informed consent was obtained from all subjects involved in the study. All facial images are anonymized and stored on local hard drives.

**Data Availability Statement:** Data are not available due to IBR requiring facial images to be stored locally.

**Acknowledgments:** The authors greatly appreciate the PRNU extraction MATLAB code provided by Jessica Fridrich at the University of Binghamton [23]. Special thanks are also given to Alexandra Taylor and Frank Lollo of the Ford Motor Company, lab member Rafi Ud Daula Refat and UM—Dearborn CECS employee Goeffrey Hosker for their continued support in the Alliance Grant.

**Conflicts of Interest:** Ali Hassani is both a graduate student at the University of Michigan—Dearborn and an employee of Ford Motor Company. This conflict of interest is reviewed by having independent principal investigators at both the university and Ford validate the deliverable.

## Appendix A. Compressed Camera Source Identification

Fundamental to this research is whether the PRNU features are sufficiently retained when applying down-sampling compression. This is accomplished using a public data-set of different camera models (Dresden [43]) and a constructed data-set from the same camera models (Ras Pi Camera). Camera source identification is evaluated when applying compression (including varying the compression order) in a zonal-expected-value fashion.

The goal is to remain competitive with state-of-the-art source identification rates, e.g., 95% to 99%, while significantly reducing run-time overhead.

*Appendix A.1. Compressed Source Identification: Different Cameras*

Worst-case conditions are selected from the results of the large-scale camera identification study from Fridrich et al. [23]. They identified that PRNU is least separable when the images analyzed are of the same scene and if they are taken using the same camera model. To provide a common reference, the Dresden camera forensics image data-set is utilized [43]. While the Dresden data-set does not have cameras of the same make, it does have several families of models. For this reason, the Nikon family of cameras is selected to reflect images of same scene with similar models. 150 images are utilized, with an 80/20 training ratio. This data-set is designated as Dresden-Nikon. The camera source identification is then evaluated using the down-sampling and compression approaches described in the methodology. The image source is then identified using peak correlation energy (PCE) in a zonal-expected-value (ZEV) fashion, for all combinations of zones, down-sampling and compression processes (indicated by "pre-compression" and "post-compression", respectively). The selected classification performance metric is the mean average precision of source identifications.

Table A1 demonstrates that the order of compression is critical in retaining identification precision. When first compressing the training image and extracting the PRNU, there is significant degradation in PCE score (resulting in poor mean average precision).

**Table A1.** Dresden-Nikon data-set identification performance.

| Compression Factor | Single Zone (% Correct) | 16 Zones (% Correct) | 100 Zones (% Correct) |
|---|---|---|---|
| Full-Scale | 100% | 100% | 100% |
| Quarter-Scale (Pre-Compression) | 71.7% | 73.3% | 77.5% |
| Sixteenth-Scale (Pre-Compression) | 38.3% | 42.5% | 45.8% |
| Quarter-Scale (Post-Compression) | 97.5% | 98.3% | 99.2% |
| Sixteenth-Scale (Post-Compression) | 97.5% | 97.5% | 97.5% |

By acquiring the PRNU first and then compressing the template, sufficient information is retained. It is noted that these results also imply that some features are necessarily lost in the challenge image, as that is compressed prior to identification for run-time purposes. Performing zonal analysis helps incrementally mitigate these losses by providing secondary distribution information to the PCE score. These benefits become more pronounced with compression. Given the simplicity of this approach, these benefits are considered a good return on computational investment.

*Appendix A.2. Compressed Source Identification: Same Camera*

The next experiment addresses using cameras of the same model. A new data-set is constructed using three identical Ras Pi cameras. That is to say, they are of the same make and model, and any variation is a result of manufacturing tolerances only. 150 photos are taken for each device under four static lighting conditions (for reference, full-scale resolution is 1920 × 1080). The performance evaluation is repeated using the Ras Pi camera data-set. Here, the inter-class separability notably decreases, as the only differences in "noiseprint" are by manufacturing tolerances (which is expected from the results of Goljan et al. [23]).

Table A2 confirms the optimal identification performance features require first acquiring the PRNU template and then compressing it. As expected, identification performance

does decrease. However, the benefits of ZEV present notably more here. Given that there are fewer differences between the cameras, the improved local sensitivity provides notable increases in precision.

**Table A2.** Raspberry Pi Camera data-set identification performance.

| Compression Factor | Single Zone (% Correct) | 16 Zones (% Correct) | 100 Zones (% Correct) |
| --- | --- | --- | --- |
| Full-Scale | 100% | 100% | 100% |
| Quarter-Scale (Pre-Compression) | 63.3% | 83.3% | 85.0% |
| Sixteenth-Scale (Pre-Compression) | 58.3% | 60.0% | 63.3% |
| Quarter-Scale (Post-Compression) | 86.7% | 100% | 100% |
| Sixteenth-Scale (Post-Compression) | 80.0% | 98.3% | 100.0% |

*Appendix A.3. Compressed Source Identification: Run-Time*

Tables A3 and A4 show the run-time and memory overheads for compressed camera source identification. This is broken out by compression factor (full-scale, quarter-scale and sixteenth-scale) versus number of zones (one, sixteen and one hundred). As expected, there are significant benefits to both metrics when using the compression methodology. It is important to highlight that these metrics are purely for source identification, and they do not take into account the resources associated with compressing the image.

**Table A3.** Source-identification run-time.

| Compression Factor | Single Zone (ms) | 16 Zones (ms) | 100 Zones (ms) |
| --- | --- | --- | --- |
| Full-Scale | 505.6 | 511.7 | 562.7 |
| Quarter-Scale | 171.9 | 185.6 | 207.8 |
| Sixteenth-Scale | 40.4 | 50.1 | 62.2 |

These results indicate that the optimal condition for source identification is to use sixteenth-scale down-sampling with 16 zones. On the target hardware, this decreases run-time from 505.6 ms to 50.1 ms, a relative reduction of 91.1%, and memory from 115.33 Mb to 8.24 Mb, a relative reduction of 92.8%. These optimized values enable real-time applications on memory-constrained hardware.

**Table A4.** Source identification consumption.

| Compression Factor | Single Zone (Mb) | 16 Zones (Mb) | 100 Zones (Mb) |
| --- | --- | --- | --- |
| Full-Scale | 115.33 | 115.33 | 115.33 |
| Quarter-Scale | 32.96 | 32.96 | 32.96 |
| Sixteenth-Scale | 8.24 | 8.24 | 8.24 |

*Appendix A.4. Discussion on Compressed Source Identification*

These experiments are designed to verify that the relevant PRNU features are well maintained when applying down-sampling. PRNU is well known for purposes of source identification [23,35]; here, it is demonstrated that compression can be used as a useful tool to speed up the algorithm while retaining sufficient reliability.

**Appendix B. Noise-Verification Framework Sensitivity Analysis**

This appendix contains the full sensitivity analysis from experiments 1 and 2.

*Appendix B.1. Face-Swap-Attack Sensitivity*

Table A5 shows the full gamut of test conditions from experiment 1 (Section 4.1). Authentic images are represented by "Authentic", manual-swap on the same camera by "Swap SC-M", manual-swap on different cameras by "Swap DC-M", AI-swap on the same camera by "Swap SC-AI" and AI-swap on different cameras by "Swap DC-AI". As mentioned in the results, there is little difference between camera swap methods.

**Table A5.** Face-swap verification performance.

| Image Tampering | FZV 1 Zone | FZV 16 Zones | FZV 100 Zones |
|---|---|---|---|
| Authentic FS | 100% | 100% | 100% |
| Swap SC-M FS | 100% | 100% | 100% |
| Swap DC-M FS | 100% | 100% | 100% |
| Swap SC-AI FS | 100% | 100% | 100% |
| Swap DC-AI FS | 100% | 100% | 100% |
| **Full-Scale Mean** | **100%** | **100%** | **100%** |
| Authentic QS | 92.5% | 92.5% | 92.5% |
| Swap SC-M QS | 100% | 82.5% | 90.0% |
| Swap DC-M QS | 100% | 92.5% | 95.0% |
| Swap SC-AI QS | 90.0% | 100% | 42.5% |
| Swap DC-AI QS | 92.5% | 100% | 47.5% |
| **Quarter-Scale Mean** | **95.0%** | **93.5% †** | **73.5%** |
| Authentic SS | 85.0% | 85.0% | 85.0% |
| Swap SC-M SS | 82.5% | 82.5% | 85.0% |
| Swap DC-M SS | 80.0% | 80.0% | 70.0% |
| Swap SC-AI SS | 77.5% | 77.5% | 27.5% |
| Swap DC-AI SS | 85.0% | 85.0% | 27.5% |
| **Sixteenth-Scale Mean** | **82.0%** | **82.0%** | **59.0%** |

† represents the optimized face-swap detection algorithm: FZV with 16 zones at quarter-scale.

*Appendix B.2. Simulated Service-Denial Sensitivity*

Table A6 shows the full test conditions from experiment 2 (Section 4.2). Authentic images are represented by "Authentic", blob-swap in the center by "Swap C"; blob-swap in the top-left by "Swap TL", blob-swap in the top-right by "Swap TF", blob-swap in the bottom-left by "Swap BL" and blob-swap in the bottom-right by "Swap BR".

**Table A6.** Simulated service-denial verification performance.

| Image Tampering | ZEV 1 Zone | ZEV 16 Zones | ZEV 100 Zones |
|---|---|---|---|
| Control FS | 100% | 100% | 100% |
| Swap C FS | 100% | 100% | 100% |
| Swap TL FS | 100% | 100% | 100% |
| Swap TR FS | 100% | 100% | 100% |
| Swap BL FS | 100% | 100% | 100% |
| Swap BR FS | 100% | 100% | 100% |
| **Full-Scale Mean** | **100%** | **100%** | **100%** |

**Table A6.** *Cont.*

| Image Tampering | ZEV 1 Zone | ZEV 16 Zones | ZEV 100 Zones |
|---|---|---|---|
| Control QS | 92.5% | 100% | 100% |
| Swap C QS | 92.0% | 100% | 100% |
| Swap TL QS | 84.0% | 100% | 100% |
| Swap TR QS | 82.5% | 100% | 100% |
| Swap BL QS | 77.0% | 96.0% | 100% |
| Swap BR QS | 76.0% | 98.0% | 100% |
| **Quarter-Scale Mean** | **84.0%** | **99.0%** | **100%** |
| Control SS | 85.0% | 92.5% | 100% |
| Swap C SS | 76.0% | 83.0% | 100% |
| Swap TL SS | 70.0% | 90.0% | 100% |
| Swap TR SS | 70.0% | 89.0% | 100% |
| Swap BL SS | 74.6% | 89.0% | 100% |
| Swap BR SS | 71.0% | 91.0% | 100% |
| **Sixteenth-Scale Mean** | **74.4%** | **89.0%** | **100% ‡** |

‡ represents the optimized service-denial detection algorithm: ZEV with 100 zones at sixteenth-scale.

The location of the blob-swap has greater impact than the size. There is minimal difference between the $\frac{1}{12}$ and $\frac{1}{50}$ image-width radii for a given location; however, the center-blob-swap verification is consistently the most precise. This variation is potentially explained by having a constant tampering score threshold over all zones. Having custom thresholds for each location could improve performance. The optimal algorithm, sixteenth-scale with 100 sub-zones, is indicated by the ‡.

## Appendix C. Face Recognition Evaluation on Swapped Faces

A relevant question is whether non-detectable face-swaps would actually spoof a face recognition system. To evaluate this, a face recognizer is constructed. This algorithm is intentionally simplistic and is constructed using MATLAB's face detector, histogram of oriented gradients feature and an SVM for trained-on-the-data-set participants.

Table A7 presents the face-swap recognition results. The results demonstrate that swaps capable of spoofing the PRNU tampering score approach are not accepted by the facial recognition model. This inherently helps mitigate the risks of low-resolution AI face-swaps not being detectable.

**Table A7.** Swapped face recognition performance.

| Swap Method | Face Classification Rate (%) | | |
|---|---|---|---|
| | Full-Scale | Quarter-Scale | Sixteenth-Scale |
| Control | 100% | 100% | 97.0% |
| Swap SC-M | 100% | 100% | 72.5% |
| Swap DC-M | 100% | 100% | 65.0% |
| Swap SC-AI | 0.0% | 0.0% | 20.0% |
| Swap DC-AI | 0.0% | 0.0% | 35.0% |

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
