# Peer review of "Efficiently Mitigating Face-Swap-Attacks: Compressed-PRNU Verification with Sub-Zones"

_technologies, doi:10.3390/technologies10020046_

Round 1
Reviewer 1 Report
The manuscript is interesting and it is easy to read. However, I found two main concerns:
1) It is not well situated in the context. I missed a background explaining where these systems are useful. For example frontiers, ABC gates, etc. And I assume that the proposal makes the checks in a static way and not in movement. I recommend a reference in this sense to consider in the manuscript: [Ortega, D., Fernández-Isabel, A., de Diego, I. M., Conde, C., & Cabello, E. (2020). Dynamic facial presentation attack detection for automated border control systems. Computers & Security, 92, 101744.]
Feel free of adding several or explaining the motivation of the proposal in detail.
2) Experiments are very weak. No comparison with the SOTA is included, so it is difficult to evaluate the quality of the proposal. A well-known dataset is also needed to implement this comparison task.
I desire the best of lucks for the authors.
Author Response
Dear Reviewer,
Thank you for your time and feedback, we found it helpful and have made improvements accordingly. Please find the attached document to thoughtfully address your concerns.

Reviewer 2 Report
This research deals with an interesting and important topic. I only have a few suggestions for further improvement of the paper. Please see my comments as follows:
The authors fail to effectively write the need of research (problem statement). In addition, research objectives are unclearly written.
Provide more details about the previous works of the extant literature, which are related this research.
Too much part of the research is covered by methods and results.
The conclusion section should be further strengthened. Provide more details about the findings and its related implications. This section should include statements about originality and value. In addition, implications are very weakly written.
This research needs a professional editing. Many sentences are written in a clumsy manner. An overall improvement of English writing is necessary.
Hope my above suggestions help the authors increase the quality of the paper.
Author Response
Dear Reviewer,
Thank you for your time and feedback, we found it helpful and have made improvements accordingly. Please find the attached document to thoughtfully address your concerns.

This manuscript is a resubmission of an earlier submission. The following is a list of the peer review reports and author responses from that submission.
Round 1
Reviewer 1 Report
The goal of the paper is to enable real-time deepfake detection by comparing PRNU-based scores of different zones in the given image.
While the general idea looks interesting, it is not clear how it relates to current methods and in my opinion, the experiments also needs to be expended and clarified.
- I would suggest adding a related work section that includes more works for deepfake detection. With more details on existing PRNU-based approaches for this task and how the proposed approach extends SOTA.
- Nguyen et al. “Deep Learning for Deepfakes Creation and Detection: A Survey” 2021
- Lugstein et al. “PRNU-based Deepfake Detection” 2021
- Moreover, some of these approaches should be used for comparison. One option might be to applied your method on a publicly-available dataset and compare the performance with the reported performances.
- More information on the used dataset/s is/are needed. For instance, how many identities are contained? (The recognition performance of 100% indicates that there are too less identities included)
- What is the reason not to use publicly-available datasets for a better comparison?
- Please provide more details on the used “face recognizer” (face recognition model). For instance, how was it trained? Why don’t a embedded face recognition solution like mobileFace etc.?
- Moreover, please report the face recognition performance in more detailed terms such as False-Non Match Rate @ False Match Rate.
- Lastly, it would be interesting to see how the proposed method would performance against other deepfake creation methods to get a better view on the generalizability of the proposed approach.
I hope the comments help to improve this work.
Reviewer 2 Report
The paper focuses on securing face-recognition perception and the topic seems interesting. The authors present their work first and then show their results with lots of experiments. Howerever, it is really hard to follow their idea for readers, which starts from the abstract to the conclusion. The organization and language confuse me a lot and I cannot understand the relations between different paragraphs in the paper. Grammar errors can easily be found. For example, These modify the camera stream to replay ... (see line 2 of page 1); A new problem, however, are .... in line 22 of page 2; This purpose of this research ... in line 83 of page 3; daata-set in line 291 of page 9; and so on. Besides, it is hard for me to find their contributions.
Thus, I don't think this manuscript is ready for publication.
Reviewer 3 Report
This paper provides a new approach to evaluate the robustness of the network, considering different amount of uncertainties on training data and testing data, while traditional approach only consider the different uncertainties on testing data. Collecting the accuracy of training data and testing data for different uncertainty pairs, come to the conclusion that the amount of uncertainty in training dataset and the accuracies on the uncertainty in testing data are interdependent. Furthermore, the model with the highest accuracy trained by the data of each uncertainty can be found under different thresholds.
However, we still need to point out some shortcomings:
- The major weakness of this paper is that it is a well-known conclusion that the training data affects the accuracy of the model in different kind of test data. Therefore, from this paper we didn’t get any unknown conclusions from this article, which means, this paper has no novelty.
- Line 342, “ is obtained at ”, the number of should be 0.068, refers to the Figure 13.
- The paragraphs from Line 226 to Line 249, the uncertainty of dataset was not written consistently. For example, in line 229, the uncertainty of training dataset was written as , but it was written as UTR in line 233.
Reviewer 4 Report
Face-recognition is becoming the go-to convenience authentication method, but it also has some security vulnerabilities. To solve the face-swap-attacks problem, the author proposes an efficient noise-verification framework. The proposed method achieves good results. However, the following are some comments to the authors to help improve this paper.
- Although the author has proved the feasibility of method via experiments, but I have some doubts about this, especially for the face-swap-attacks of the same camera model, because so far, researchers can not effectively distinguish PRUN from the same camera model.
- The innovation of this paper is general. The main contribution is to apply PRNU to face recognition and uses deviation in the expected PRUN to detect and localize tampered segments.
- Although the proposed method has done a lot of experiments on public data sets, it has not simulated the actual face recognition process for method verification.
- The author should provide detailed analysis about the low pass cutoff and high pass cutoff of tamping filter in the experimental section to verify the robustness of tamping filter.